# Associations between Clinicopathological Characteristics and Intraoperative Opioid Requirements during Endoscopic Submucosal Dissection with Monitored Anesthesia Care: A Retrospective Study

**DOI:** 10.3390/jcm13113119

**Published:** 2024-05-26

**Authors:** Hyun Il Kim, Da Hyun Jung, Sung Jin Lee, Yong Chan Lee, Sang Kil Lee, Ga Hee Kim, Ho Jae Nam, Sihak Lee, Hyo-Jin Byon, Sung Kwan Shin

**Affiliations:** 1Department of Anesthesiology and Pain Medicine, Severance Hospital, Yonsei University College of Medicine, Seoul 03722, Republic of Korea; choco8926@yuhs.ac (H.I.K.); sj122345@yuhs.ac (S.J.L.); hisjourney@yuhs.ac (H.J.N.); 2Division of Gastroenterology, Department of Internal Medicine, Institute of Gastroenterology, Severance Hospital, Yonsei University College of Medicine, Seoul 03722, Republic of Korea; jungdh@yuhs.ac (D.H.J.); leeyc@yuhs.ac (Y.C.L.); sklee@yuhs.ac (S.K.L.); smallgh@yuhs.ac (G.H.K.); 3Stanley Center for Psychiatric Research, Broad Institute of Massachusetts Institute of Technology and Harvard, Cambridge, MA 02142, USA; leesihak@broadinstitute.org

**Keywords:** endoscopic submucosal dissection, gastric cancer, monitored anesthesia care, opioid

## Abstract

**Background and study aims:** Endoscopic submucosal dissection is used to treat early gastric neoplasms. Compared with other endoscopic procedures, it requires higher doses of opioids, leading to adverse events during monitored anesthesia care. We investigated the correlations between clinicopathological characteristics and intraprocedural opioid requirements in patients who underwent endoscopic submucosal dissection under monitored anesthesia care. **Patients and methods:** The medical records of patients who underwent endoscopic submucosal dissection under monitored anesthesia care were retrospectively reviewed. The dependent variable was the total dose of fentanyl administered during the dissection, while independent variables were patient demographics, the American Society of Anesthesiologists physical status classification, preoperative vital sign data, and the pathological characteristics of the neoplasm. Correlations between variables were examined using multiple regression analysis. **Results:** The study included 743 patients. The median total fentanyl dose was 100 mcg. Younger age (coefficient −1.37; 95% confidence interval [CI] −1.78 to −0.95), male sex (16.12; 95% CI 6.99–25.24), baseline diastolic blood pressure (0.44; 95% CI 0.04–0.85), neoplasm length (1.63; 95% CI 0.90–2.36), and fibrosis (28.59; 95% CI 17.77–39.42) were positively correlated with the total fentanyl dose. Total fentanyl dose was higher in the differentiated (16.37; 95% CI 6.40–26.35) and undifferentiated cancers group (32.53; 95% CI 16.95–48.11) than in the dysplasia group; no significant differences were observed among the others. The mid-anterior wall (22.69; 95% CI 1.25–44.13), mid-posterior wall (29.65; 95% CI 14.39–44.91), mid-greater curvature (28.77; 95% CI 8.56–48.98), and upper groups (30.06; 95% CI 5.01–55.12) had higher total fentanyl doses than the lower group, whereas doses did not significantly differ for the mid-lesser curvature group. **Conclusions:** We identified variables that influenced opioid requirements during monitored anesthesia care for endoscopic submucosal dissection. These may help predict the needed opioid doses and identify factors affecting intraprocedural opioid requirements.

## 1. Introduction

Gastrectomy via open or laparoscopic surgery is the traditional treatment for gastric neoplasms; however, endoscopic submucosal dissection (ESD) is increasingly being used because it can provide definitive treatment with fewer complications than those associated with surgery [1,2,3]. Rather than under general anesthesia, endoscopic procedures are typically performed under monitored anesthesia care (MAC), which has both advantages and disadvantages. Patients treated under MAC showed improved recovery profiles compared to those treated under general or regional anesthesia [4]. On the other hand, the risk of hypoxia is higher in MAC, which does not involve invasive airway management (i.e., intubation) [5]. In endoscopic procedures, the scope is placed in the patient’s mouth, making it difficult to quickly manage the airway in the event of respiratory depression or hypoxia [6,7].

ESD procedures require higher doses of opioids than those required for conventional endoscopic procedures owing to the severity of pain during lesion ablation; these higher doses can lead to serious adverse effects such as respiratory depression or hypoxia in patients undergoing ESD [8,9]. If it were possible to predict adequate opioid requirements for patients with ESD, adverse effects caused by opioid overdose could be prevented. It would also help procedures go smoothly by enabling the provision of ideal doses for analgesia. To date, several studies have examined pain after ESD procedures, showing that the severity of postoperative pain appears to vary depending on the clinicopathological characteristics of the patients. However, to the best of our knowledge, no study has focused on the opioid requirements for the procedure under MAC [10,11,12,13,14]. Therefore, we aimed to investigate the correlation between clinicopathological characteristics and intraprocedural opioid requirements in patients who underwent ESD with MAC.

## 2. Material and Methods

### 2.1. Patients

This retrospective cohort study was approved by the Institutional Review Board of Severance Hospital, Yonsei University Health System (approval number: 4-2022-1431, date of approval: 6 April 2023). The study design followed STROBE guidelines. Informed consent was waived because of the retrospective nature of the research, and the research involved no more than minimal risk to the patients. Data were collected by retrospectively examining the medical records of patients aged 18 years or older who underwent ESD with MAC at the single tertiary referral hospital between 1 February 2022, and 3 April 2023. Patients who received remimazolam and fentanyl, the standard MAC regimens for ESD procedures performed at the institution, were included in the study. The exclusion criteria were as follows: (1) patients who underwent general anesthesia rather than MAC, (2) those who had two or more neoplasms resected in 1 day, and (3) those with missing values in the medical record.

### 2.2. Procedure

The standard sedation protocol was as follows: When the patient entered the operating room, blood pressure, heart rate, and pulse oximetry were monitored. An intravenous (IV) bolus of glycopyrrolate 0.1 mg was administered as premedication. Preoxygenation was performed. A remimazolam 5 mg IV bolus was administered in 1 min, and the dosage was maintained at an IV infusion rate of 10–20 mg/h^−1^ during the procedure. The level of sedation was adjusted to 2–3 points on the Modified Observer’s Assessment of Alertness/Sedation Scale (MOAA/S). At the beginning of the procedure, a fentanyl 50 mcg IV bolus was routinely administered, along with a bolus of remimazolam, to prevent painful stimuli during endoscope insertion. During the procedure, an additional 25 mcg IV bolus of fentanyl was administered if the patient awakened due to a painful stimulus. If hypoxia due to respiratory depression occurred during the procedure, the remimazolam infusion rate was reduced, and the jaw-thrust maneuver was performed to secure the airway. After the procedure was completed, a flumazenil 0.3 mg IV bolus was administered to reverse the sedation effect of remimazolam, and the patient was moved to the postanesthetic care unit.

When the induction agent was injected, an endoscope was inserted, and the location of the lesion was confirmed and marked. Submucosal injections of a solution consisting of epinephrine (0.01 mg/mL) and 0.8% indigo carmine were administered, and a mucosal incision was performed, followed by submucosal layer dissection. A dual knife (KD-650Q; Olympus, Tokyo, Japan) or an insulated-tip knife (KD-610L; Olympus Optical, Tokyo, Japan) was used for incision and dissection of the lesion. After ESD, the bleeding site was ablated, and the procedure was completed.

### 2.3. Data Analysis

The dependent variable was the total dose of fentanyl administered for ESD, and the independent variables included in the model for analysis were as follows: patient demographics, American Society of Anesthesiologists Physical Status (ASA-PS) classification, preoperative vital sign data (baseline blood pressure, heart rate), and pathological characteristics (location of the lesion, fibrosis, length of the lesion measured by endoscopy, histology type). The location of the lesion was subdivided into six groups as follows: upper (cardia, fundus), mid (divided into four parts: anterior wall [mid-AW], posterior wall [mid-PW], greater curvature [mid-GC], and lesser curvature [mid-LC]), and lower (antrum, pre-pylorus). Endoscopists confirmed fibrosis during the procedure. The histology of the neoplasm was confirmed by pathology and categorized as follows: dysplasia (epithelial dysplasia, low- or high-grade), differentiated cancer (tubular adenocarcinoma, well or moderately differentiated), undifferentiated cancer (tubular adenocarcinoma, poorly differentiated; gastric carcinoma, poorly cohesive or signet ring cell type), or other (neuroendocrine tumor, etc.) Statistical analyses were then performed; each independent variable was compared to the dependent variable using univariate linear regression analyses for dichotomous and continuous variables and Kruskal–Wallis tests for categorical variables. Independent variables showing statistical significance were considered potentially significant confounding variables, and multiple linear regression analysis was fitted with significant independent variables. Variable selection was performed through backward elimination based on the Akaike Information Criteria, and variables showing statistical significance in the final model were considered the final results. Statistical significance was set at *p* < 0.05, and all statistical analyses were performed by the authors using R software (version 4.1.3, R Core Team, R Foundation for Statistical Computing, Vienna, Austria).

## 3. Results

In total, 799 patients were enrolled during the screening period. After excluding 56 patients who met the exclusion criteria, the data from 743 patients were used in the data analysis. Figure 1 shows a flow diagram of the study. The median total fentanyl dose was 100 mcg (interquartile range [IQR], 100–150). Table 1 shows the clinicopathological features of the patients. The median age was 66 years (IQR, 60–74), and 243 (33%) patients were women. Most patients were in ASA-PS class II (I: 56, 8%; II: 611, 82%; III: 75, 10%; IV: 1, 0%). The gastric neoplasms were mostly located in the lower group (69%). The histological groups according to the final pathology included 467 (63%) patients with dysplasia, 191 (26%) with differentiated cancer, 71 (10%) with undifferentiated cancer, and 14 (2%) with “other”. Most pathological neoplasms in the “other” group were neuroendocrine tumors.

Table 2 shows the results of the univariate linear regression analyses for each independent variable. The following variables showed statistical significance: age, sex, height, weight, number of previous procedures, diastolic blood pressure (baseline), fibrosis, length of the neoplasm (measured by endoscopy), histology, and location. Multiple linear regression analyses were performed, including all variables as independent variables. The following variables were excluded from the final results through variable selection: height, weight, and number of previous procedures.

Table 3 shows the results of the multiple linear regression analyses. There was a negative association between total fentanyl dose and age (coefficient, −1.37; 95% confidence interval [CI], −1.78 to −0.95; *p* < 0.001), and men received more total fentanyl (coefficient, 16.12; 95% CI, 6.99–25.24; *p*-value = 0.001). Baseline diastolic blood pressure (coefficient, 0.44; 95% CI, 0.04–0.85; *p*-value = 0.031), length of neoplasm (coefficient, 1.63; 95% CI, 0.90–2.36; *p*-value < 0.001), and fibrosis (coefficient, 28.59; 95% CI, 17.77–39.42; *p*-value < 0.001) were positively correlated with total fentanyl dose. For histology, using the dysplasia group as a reference, the total fentanyl dose was higher for those with differentiated cancer (coefficient, 16.37; 95% CI, 6.40–26.35; *p*-value = 0.001) and undifferentiated cancer (coefficient, 32.53; 95% CI, 16.95–48.11; *p*-value < 0.001); there were no significant differences for the other groups. For tumor location, using the lower group as a reference, the mid-AW (coefficient, 22.69; 95% CI, 1.25–44.13; *p*-value = 0.038), mid-PW (coefficient, 29.65; 95% CI, 14.39–44.91; *p*-value < 0.001), mid-GC (coefficient, 28.77; 95% CI, 8.56–48.98; *p*-value = 0.005), and upper (coefficient, 30.06; 95% CI, 5.01–55.12; *p*-value = 0.019) groups had higher total fentanyl doses, while there was no significant difference for the mid-LC group.

## 4. Discussion

In this study, we identified the characteristics of patients who underwent ESD with MAC that affected the total fentanyl dose required for the procedure. Younger age and male sex were associated with higher fentanyl doses. The lower group (antrum and pre-pylorus) required the lowest dose of fentanyl, and histological types of cancer required more fentanyl than that required for dysplasia. The presence of fibrosis and a longer lesion length (as measured by endoscopy) were associated with higher fentanyl doses. These results can be used to predict the opioid dose required for ESD; the risk of opioid overdose can be recognized, and the proper precautions taken during ESD procedures in patients with the aforementioned characteristics.

A physician administering sedation should titrate the doses of sedative agents and analgesics according to the features of the procedure. Owing to the nature of ESD procedures, which require extensive lesion ablation in the submucosal layer, insufficient analgesia induces patient movement and makes the procedure difficult, whereas doses that are too high can cause adverse events such as respiratory distress or hypotension. Therefore, it is important to predict the ideal dose of analgesics, and this varies among patients. At our institution, MAC with remimazolam and fentanyl is the standard protocol for ESD sedation. Remimazolam is associated with a lower risk of respiratory depression and hemodynamic instability than previously used anesthetics such as propofol and midazolam [15,16,17]. However, there is always the possibility that a patient may be at risk of respiratory distress due to excessive doses of opioids [18]. Therefore, administering an excessive dose of opioids such as fentanyl poses a risk to patient safety and should be avoided. The clinicopathological characteristics identified in this study that influence opioid requirements may help predict the appropriate opioid requirements for patients undergoing ESD procedures, reducing the risk of adverse events associated with excessive opioid administration and simplifying the procedure.

According to our results, the total fentanyl dose varied depending on the location of the lesion. In particular, lesions in the lower group (antrum and pre-pylorus) required a lower opioid dose during the procedure than that required by groups with lesions in other locations. These results may be due to differences in procedure difficulty depending on the location of the lesion. In previous studies, tumors located in the upper or middle parts of the stomach required significantly longer ESD procedure times than those located in the lower stomach [19,20]. The technical difficulty of ESD procedures varies in relation to the characteristics of the lesion. When lesions involve the gastric folds, proper positioning of the endoscope is difficult [21]. Furthermore, when bleeding occurs, it is difficult to gain visibility for the procedure because pooled blood disturbs the endoscopic hemostasis. Arterial bleeding occurs more frequently with lesions in the middle and upper thirds of the stomach than with lesions in the lower third [19]. Therefore, it is expected that the difficulty of the procedure, and therefore the opioid requirement for sedation, will be greater for tumors in locations other than the lower third of the stomach; for these lesions, it is necessary to prepare for the risk of adverse events due to excessive doses of opioids.

We confirmed that the pathological characteristics of gastric cancer are associated with opioid requirements during ESD under MAC. The more severe the fibrosis and the longer the lesion, the greater the opioid requirement during the procedure. The following can be expected: the more severe the fibrosis of the lesion, the greater the amount of solution injected into the submucosal layer and the greater the force required to resect the lesion with an electrosurgical knife. It can also be expected that the longer the lesion, the more extensive the required lesion ablation will be; therefore, the opioid requirement will be greater. Regarding the histologic type, the cancer group was associated with higher opioid requirements than the dysplasia group. Theoretically, compared with dysplasia, cancer typically has a deeper invasion depth and more angiogenesis, leading to more coagulation with the electrosurgical knife [22]. This could lead to more pain during the procedure and increased opioid requirements for ESD. Therefore, the pathological characteristics of gastric cancer should be considered when predicting opioid requirements in patients undergoing ESD with MAC. If the lesion is large, fibrosis is severe, and high-grade malignancy is suspected, opioid requirements will likely be great, and the risk of opioid overdose should be anticipated.

It is natural to expect that as weight and height increase, the total fentanyl requirement for successful analgesia will also increase. In our study, both weight and height were found to be significantly associated with total fentanyl dose when analyzed using univariate linear regression analysis. However, in the multivariate linear regression analysis, which included all other significant variables, both weight and height were excluded during the variable selection process and were not included in the final model. This result suggests that weight and height were not independently influencing the total fentanyl dose but were either indirectly correlated with other covariates, or their influence was weaker than that of the other variables included in the final model. We believe this indicates the clinical implications of our findings, as it suggests that the variables ultimately included in the multivariate regression model had a greater impact on the total fentanyl dose required during the procedure than weight and height.

Opioid requirements were found to decrease with age. This finding is consistent with those of previous studies in which increasing age led to changes in pharmacokinetics and pharmacodynamics, resulting in increased susceptibility to opioids in older patients [23,24]. Similarly, women were observed to have reduced opioid requirements; this is consistent with the results of previous studies, in which women had a higher pain tolerance and required less morphine for postoperative analgesia [23]. In the present study, a higher baseline diastolic blood pressure was associated with an increase in the total required fentanyl dose, and we are not aware of any previous studies with similar results. Anxiety increases blood pressure and activates the sympathetic nervous system, which, in turn, amplifies pain [25,26]. It is possible that patients in this study who were anxious before the procedure had increased baseline blood pressures and increased pain susceptibility, resulting in increased opioid requirements. Additional well-designed studies are required to confirm this hypothesis.

This study had some limitations. First, it was a retrospective study, which means it was prone to bias owing to confounding variables. To minimize the risk of bias inherent in retrospective studies, we performed a multivariate regression analysis including all potential confounding variables. Second, this study was conducted at a single institution. Careful consideration should be given to extrapolating our findings to other populations. Finally, the agents used in this study were limited to remimazolam and fentanyl, which are the standard regimens for ESD sedation at our institution. As mentioned above, remimazolam induces fewer hemodynamic changes and respiratory disturbances in patients compared to other sedatives, such as propofol and midazolam. The results may differ if sedation is performed with other agents.

## 5. Conclusions

We identified variables that influenced opioid requirements during MAC for ESD. These results could be used to predict the doses of opioids needed during ESD and to identify factors that affect intraprocedural opioid requirements.

## Figures and Tables

**Figure 1 jcm-13-03119-f001:**
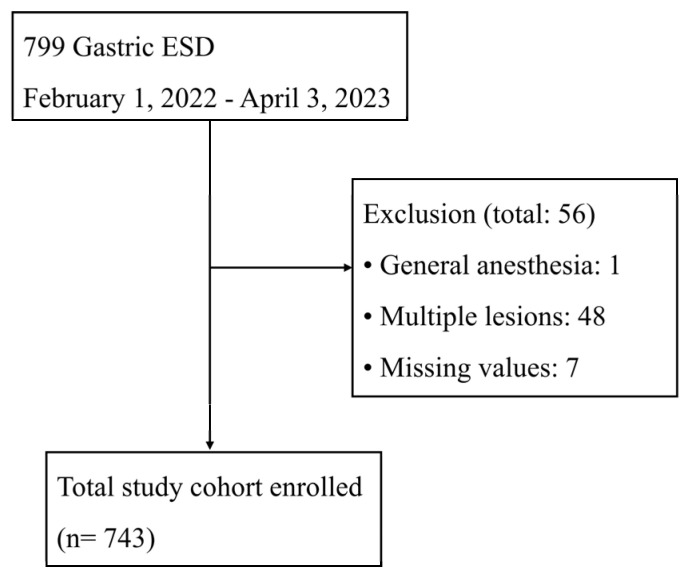
Flow diagram of the study. ESD: endoscopic submucosal dissection.

**Table 1 jcm-13-03119-t001:** Patient characteristics and perioperative variables.

Patient Characteristics	
Age (years)	66 (60–74)
Male sex	500 (67%)
Height (cm)	165.2 (158.8–171.0)
Weight (kg)	65.9 (58.0–73.6)
History of previous gastric surgery	10 (1%)
ASA-PS class	
1	56 (8%)
2	611 (82%)
3	75 (10%)
4	1 (0%)
Number of previous procedures	
0	657 (88%)
1	64 (9%)
2	18 (2%)
3	3 (0%)
4	1 (0%)
Perioperative data	
Systolic blood pressure (mmHg, baseline)	137 (125–153)
Diastolic blood pressure (mmHg, baseline)	79 (71–86)
Heart rate (bpm, baseline)	81 (72–91)
Fibrosis	160 (22%)
Length of neoplasm measured by endoscopy (mm)	15 (10–17)
Histology	
Dysplasia	467 (63%)
Differentiated cancer	191 (26%)
Undifferentiated cancer	71 (10%)
Other	14 (2%)
Location	
Lower	513 (69%)
Mid-AW	31 (4%)
Mid-PW	65 (9%)
Mid-GC	37 (5%)
Mid-LC	75 (10%)
Upper	22 (3%)

Values are presented as medians (interquartile ranges) or numbers (proportions). ASA-PS: American Society of Anesthesiologists Physical Status, Mid-AW: mid-anterior wall, Mid-PW: mid-posterior wall, Mid-GC: mid-greater curvature, Mid-LC: mid-lesser curvature.

**Table 2 jcm-13-03119-t002:** Univariate linear regression analysis.

Variable	Coefficient	*p*-Value
Age (years)	−1.41	<0.001
Male sex	14.99	0.003
Height (cm)	1.28	<0.001
Weight (kg)	0.89	<0.001
History of previous gastric surgery	−5.03	0.809
ASA-PS class	−0.48	0.933
Number of previous procedures	−10.73	0.037
Perioperative data		
Systolic blood pressure (mmHg, baseline)	−0.22	0.093
Diastolic blood pressure (mmHg, baseline)	0.59	0.009
Heart rate (bpm, baseline)	0.12	0.081
Fibrosis	39.23	<0.001
Length of neoplasm measured by endoscopy (mm)	1.63	<0.001
Histology ^a^		<0.001
Location ^a^		<0.001

^a^ Variables were considered categorical, and *p*-values were calculated using the Kruskal–Wallis test. ASA-PS: American Society of Anesthesiologists Physical Status.

**Table 3 jcm-13-03119-t003:** Multivariate linear regression analysis.

Variable	Coefficient	95% Confidence Interval	*p*-Value
Age	−1.37	−1.78, −0.95	<0.001
Male sex	16.12	6.99, 25.24	0.001
Diastolic blood pressure (mmHg, baseline)	0.44	0.04, 0.85	0.031
Length of neoplasm measured by endoscopy (mm)	1.63	0.90, 2.36	<0.001
Fibrosis	28.59	17.77, 39.42	<0.001
Histology			
Dysplasia (reference)	–	–	–
Differentiated cancer	16.37	6.40, 26.35	0.001
Undifferentiated cancer	32.53	16.95, 48.11	<0.001
Other	2.40	−30.00, 34.80	0.884
Location			
Lower (reference)	–	–	–
Mid-AW	22.69	1.25, 44.13	0.038
Mid-PW	29.65	14.39, 44.91	<0.001
Mid-GC	28.77	8.56, 48.98	0.005
Mid-LC	12.22	−2.00, 26.44	0.092
Upper	30.06	5.01, 55.12	0.019

Mid-AW: mid-anterior wall, Mid-PW: mid-posterior wall, Mid-GC: mid-greater curvature, Mid-LC: mid-lesser curvature.

## Data Availability

The data presented in this study are available on request from the corresponding author due to the information privacy of the patients.

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
