# Peer review of "Associations between Clinicopathological Characteristics and Intraoperative Opioid Requirements during Endoscopic Submucosal Dissection with Monitored Anesthesia Care: A Retrospective Study"

_jcm, 2024, doi:10.3390/jcm13113119_

Round 1

Reviewer 1 Report

Comments and Suggestions for Authors

This study is well done, the methodology is sound and the conclusions are pertinent, supported by the data. The only comment I have is that the anesthetic protocol seems particular. Premedication by glycorpyrrolate is not a usual practice, as is the administration of remimazolam and fentanyl. typically used drugs would be short acting sedative and analgesic agents, such as propofol or remifentanil. could the authors comment on the choice of these drugs? the use of remimazolam limits the interest to many anesthesiologists, as this drug is not yet licensed in many countries. 

Author Response

 Thank you very much for your valuable comment. We routinely use glycopyrrolate in patients undergoing ESD because it suppresses salivation, making it easier for the endoscopist to perform the procedure. Since glycopyrrolate is an anti-muscarinic agent with no sedative effect, we do not believe it would have affected the primary endpoint of this study.

 At our institution, we use fentanyl instead of remifentanil as the standard opioid for ESD procedures. The unpredictable nature of ESD procedures makes it difficult to estimate their duration. For most shorter cases, an initial bolus of fentanyl provides adequate analgesia. Additionally, depending on the pathology or anatomical location of the tumor, there may be sudden onset of intense pain during the procedure. In such cases, an additional bolus of fentanyl can effectively manage the pain. Administering a bolus of remifentanil in response to sudden, severe pain during the procedure is likely to cause apnea. For these reasons, we prefer fentanyl over remifentanil as our standard opioid.

 Currently, remimazolam is routinely used in our center for ESD patients. Remimazolam offers the advantage of causing fewer hemodynamic changes and respiratory disturbances in patients compared to propofol and midazolam, and it can be quickly reversed with flumazenil. Due to the nature of endoscopic procedures, where the scope is introduced through the oral cavity, rapid airway securement can be challenging in the event of respiratory distress. We believe that the benefits of remimazolam mentioned above will help reduce the risks associated with these endoscopic procedures. Additionally, because it is less likely to provide an analgesic effect while maintaining patient sedation, we believe it may have a minimal impact on the total opioid dose required to control painful stimuli during the procedure, which was the goal of the study.

However, as you have pointed out, this study only included patients who received MAC with the specific agents that constitute the standard anesthetic regimen in our institution. Therefore, extrapolating the results of this study to patients receiving other agents would require caution. We acknowledge your point and have incorporated an additional statement in the discussion section of the manuscript. Once again, we appreciate your valuable comments.

Reviewer 2 Report

Comments and Suggestions for Authors

I read with great interest the manuscript by Kim et al. on the associations between clinico-pathological characteristics and intraoperative opioid requirements during endoscopic submucosal dissection with monitored anesthesia care. The manuscript is sound and well written. However, I have some comments to make:

- Line 71. Please provide the date of ethical committee approval.

- Line 86. Please remove "and".

- Results. I believe authors should also report the incidence of adverse events such as hypoxia, respiratory depression or need to airway management in their population. In fact, it seems rather obvious that higher weight or height is related to the need to higher dosage of fentanyl to achieve the same analgesic result. However, this does not necessarily implicate that higher doses of fentanyl in those patients lead to higher incidence of adverse events. Please clarify this concept in the discussion.

Author Response

 Thank you very much for your kind and valuable comments. We have added the date of approval and removed “and” from Line 86 as you pointed out.

 Our protocol of using remimazolam and fentanyl was associated with a very low incidence of hypoxia. No adverse events of clinically significant hypoxia, respiratory depression, or airway management issues were recorded. Only four patients out of a total of 743 had pulse oximetry readings below 92% for more than five minutes, as noted on the anesthesia record sheet. This is likely because, in cases of hypoventilation, patients can be immediately awakened to stimulate respiration, and flumazenil can be administered if the depth of sedation is excessive. However, these events were not recorded, and the retrospective nature of the study precludes accurate data collection. We would be happy to include above explanation in the discussion if needed.

 As you pointed out, it is natural to expect that as weight and height increase, the total fentanyl requirement for successful analgesia would also increase. In our study, both weight and height were found to be significantly associated with total fentanyl dose when analyzed using univariate linear regression analysis. However, in the multivariate linear regression analysis, which included all other significant variables, both weight and height were excluded during the variable selection process and were not included in the final model. This result suggests that weight and height were not independently influencing the total fentanyl dose but were either indirectly correlated with other covariates, or their influence was weaker than that of the other variables included in the final model. We believe this indicates the clinical implications of our findings, as it suggests that the variables ultimately included in the multivariate regression model had a greater impact on the total fentanyl dose required during the procedure than weight and height. We have added a mention of this to the discussion in the main text. Once again, thank you for your valuable comments.